# Child Growth Curves in High-Altitude Ladakh: Results from a Cohort Study

**DOI:** 10.3390/ijerph17103652

**Published:** 2020-05-22

**Authors:** Wen-Chien Yang, Chun-Min Fu, Bo-Wei Su, Chung-Mei Ouyang, Kuen-Cheh Yang

**Affiliations:** 1Department of Pediatrics, National Taiwan University Hospital Hsin-Chu Branch, Hsinchu 30059, Taiwan; wenchienyang@gmail.com (W.-C.Y.); b87401039@ntu.edu.tw (C.-M.F.); 2Department of Family Medicine, National Taiwan University Hospital, Taipei 100225, Taiwan; b99401090@ntu.edu.tw; 3Department of Dietetics, National Taiwan University Hospital Hsin-Chu Branch, Hsinchu 30059, Taiwan; maggieoy@gmail.com; 4Department of Family Medicine, National Taiwan University Hospital Bei-Hu Branch, Taipei 108206, Taiwan

**Keywords:** child growth, growth curve, high altitude, body weight, body height

## Abstract

High prevalence of child underweight and stunting in high-altitude areas has often been reported. However, most previous studies on this topic were cross-sectional. Another critical concern is that using the World Health Organization (WHO) Child Growth Standards to evaluate child growth in high-altitude areas may lead to overestimations of underweight and stunting. Our study aimed to evaluate the long-term growth pattern of children (3 to 18 years) above the altitude of 3500 m in Ladakh, India. The participants’ body weight (BW), body height (BH), and body mass index (BMI) were measured annually according to the WHO Child Growth Standards for children under 5 years old and the WHO reference data for children aged 5 to 19 years. The generalized estimating equation (GEE) was used to estimate the means and *z*-scores of BW, BH, and BMI at different ages. A total of 401 children were enrolled from 2012 to 2018. Their mean *z*-scores of BW, BH, and BMI were −1.47, −1.44, and −0.85 in 2012 and increased to −0.74, −0.92, and −0.63 in 2018. This population’s specific growth curve was also depicted, which generally fell below the 85th percentile of the WHO standards. This is the first cohort study about long-term child growth patterns in a high-altitude area. The detailed underlying mechanisms of our findings need future research on more representative data of high-altitude populations.

## 1. Introduction

Living in high-altitude areas affects child health through various mechanisms. Low-oxygen environments, scarceness of resources, and lack of health infrastructure are common risk factors of poor health and restricted growth [1]. Children are more likely to have respiratory infectious diseases, poor nutritious condition, and growth retardation [2,3]. Some of these risk factors are inherently associated. Although there is no single leading risk factor, the impact of living in high-altitude areas on child health is certain.

High prevalence of child underweight and stunting in high-altitude areas has often been reported [4,5,6,7,8,9,10,11,12,13,14,15]. Children from Peruvian Quechua [7,8], Bolivian Aymara [9,10], and Tibet were generally shorter and lighter than their low-altitudinal counterparts of the same ethnicities [5,6,11,12]. From studies conducted in Ladakh, India, Wilson et al. showed that children under the age of 8 years were shorter and lighter compared to the international reference population [13]. Moreover, children in high-altitude areas tended to have delayed or ill-defined growth spurts as they reached adolescence [14,15].

However, there remain two critical concerns about previous studies regarding this topic. First, most of them were cross-sectional studies without longitudinal observations. This limitation hindered researchers in understanding long-term child growth trends in high-altitude areas and developing age-appropriate interventions that match their health needs [12]. Second, most previous studies applied the international WHO Child Growth Standards as the reference to evaluate child and adolescent growth. However, this approach has been deeply criticized [16,17]. Overestimating the prevalence of underweight and stunting was expected, given that the WHO standards were study results from lowland-living populations in six countries [18]. As a result, ongoing research asked for the development of region-specific or ethnicity-specific growth curves for children from diverse environments [19,20,21].

To address these issues, we conducted a cohort study to investigate the long-term growth pattern for a group of children in high-altitude Ladakh, India, and aimed to present their growth curves by following them for six consecutive years.

## 2. Materials and Methods

### 2.1. Ladakh Overview

Our partner institution is located in Leh (3500 m), a town in Leh district of the union territory of Ladakh, India. Based on the 2011 India census, Leh district had a population of 133,487 with the lowest density in India, 3 persons per square kilometer [22,23]. The inhabitants in Leh district include Indo-Aryan people and immigrants from Tibet and Skardu [23]. Although Leh district is covered by deserts and has extreme climate conditions, subsistence agriculture is the major economic activity [23,24]. Tourism has become a growing industry in Ladakh [23]. Ladakh used to belong to the state of Jammu and Kashmir, which had a per capita income of Rs. 94,992 (USD 1400). The number was less than the overall Indian average of Rs. 126,406 (USD 1700–1800) [25,26]. The Human Development Index of Jammu and Kashmir was 0.679 in 2017, which was at the medium level of human development [27].

### 2.2. Participants

This is a longitudinal follow-up study from 2012 to 2018. All of the study participants were students from our partner school, a private charity institution founded and supported by the Dalai Lama Trust. This boarding school started to recruit students in kindergarten and low grades in 2008. In 2018, the school had about 322 students from kindergarten to the 10th grade. Most of the students are descendants of Indo-Aryans living in remote areas of Ladakh, including Hanu (3160 m), Dah (2800 m), Batalik (2800 m), and Garkon (3700 m). According to the school’s enrollment policy, it recruited students from socioeconomically disadvantaged backgrounds such as orphans, students with a single parent, or students from families that cannot afford education. All of the students in our partner school were eligible for the study and were included in this study.

### 2.3. Health Examination

The National Taiwan University College of Medicine has deployed a volunteer service team to Ladakh annually since 2012. The team was composed of medical students, nurses, and physicians with expertise in family medicine, infectious diseases, pediatrics, dentistry, etc. Our team collaborated with the partner school and conducted this longitudinal study to measure students’ health conditions from 2012 to 2018. All of the students accepted health examination unless the student was absent on the examination day. The annual examination dates were 7 August 2012, 27 July 2013, 6 August 2014, 3 August 2015, 8 August 2016, 7 August 2017, and 11 August 2018. The volunteer physicians performed physical examinations covering ear, throat, heart, chest, abdomen, skin, past medical history, and menstruation history for girls. All volunteers were well-trained through standardized courses and responsible for eye health, color blindness tests, and anthropometric measurements, which included body weight (BW) in kg, body height (BH) in cm, and body mass index (BMI) in kg/m^2^. In addition, volunteer dentists started to perform full mouth dental caries detection for adolescents aged over 12 years according to the WHO standard guideline from 2017 [28]. The examination of stool parasites and ova was initiated in 2018 by an experienced technician and an infectious disease specialist. We obtained informed consent from our partner school to do the analyses. This study was also approved by the institutional ethics committee at the National Taiwan University Hospital (No.201904055RINA).

### 2.4. Growth Evaluation

We used the WHO Child Growth Standards for children under 5 years old and the WHO reference data for children aged 5 to 19 years as international standards [29,30]. We made series plots of children’s BW, BH, and BMI by age using 50 participants randomly sampled to show the individual growth trend superimposed on the WHO standards [29,30]. Underweight, stunting, and wasting were defined by a *z*-score lower than −2 of BW, BH, and BMI, respectively. This population’s growth curves of BW, BH, and BMI were presented by showing the 3rd, the 15th, the 50th, the 85th, and the 97th percentiles according to the distribution of participants at different ages in months. The growth rate of body height was defined as body height increment (cm) within a year; growth spurts were identified as the times with the most rapid growth rates of body height in a year (cm/year).

### 2.5. Statistical Analysis

The means and standard errors (SE) of BW, BH, and BMI were calculated. As BW, BH, and BMI were measured annually in this longitudinal study, generalized estimating equation (GEE) was used to reduce bias coming from missing values and repeated measurements and to estimate the means of participants’ anthropometric measurements at each age in months [31]. In our GEE model, BW, BH, and BMI are dependent variables, and ages in months are independent variables. The first degree autoregressive was applied to the working correlation matrix. Growth rates of body height for both genders were calculated at different ages, and the results were plotted.

Each participant’s *z*-scores of BW, BH, and BMI were calculated on the basis of the WHO standards using SAS MACRO codes from WHO websites [32,33]. This population’s averages of *z*-scores from 2012 to 2018 were calculated. All the analyses and plots of this study were generated using SAS software version 9.3, Copyright (c) 2002–2010 by SAS Institute Inc., Cary, NC, USA.

## 3. Results

During the period from 2012 to 2018, there were 401 children and adolescents aged 3 to 18 years enrolled in this study, including 206 boys (51.4%) and 195 girls (48.6%). New students entered this cohort annually; some might have left due to personal reasons, and some graduated in 2018. By year, there were 195 participants examined in 2012, 216 in 2013, 242 in 2014, 280 in 2015, 290 in 2016, 301 in 2017, and 322 in 2018. The rates of loss of follow-up were 0%, 0.5%, 2.9%, 11.8%, 9.0%, and 9.6% from 2013 to 2018. The details about the numbers of participants assessed each year and rates of loss of follow-up were shown in Table 1. At the time of each checkup, 34.0% of participants were under the age of 5 years, 32.5% were between 5 and 10 years old, 28.2% were between 10 and 15 years old, and 5.3% were above 15 years old. A total of 145 participants received complete follow-up for six consecutive years, and 43 participants were followed for five consecutive years. The mean and median follow-up periods were 3.7 and 4 years.

Table 2 showed the means of BW, BH, and BMI at different ages for both genders by GEE. The serial BW, BH, and BMI of 50 random samples, which were selected with equal probability without replacement from the original population, were presented superimposed on the WHO standards in Figure 1. Most of the values increased with age. During the period from 2012 to 2018, the percentages of participants who were underweight and stunted decreased yearly. For underweight, the serial percentages were 46.7%, 30.8%, 26.5%, 18.2%, 10.3%, 8.1%, and 3.6%. For stunting, the serial percentages were 44.6%, 40.7%, 33.3%, 27.5%, 26.9%, 24.3%, and 18.0%. For wasting, the serial percentages were 14.4%, 9.7%, 10.2%, 6.9%, 7.2%, 8.0%, and 9.0%. The mean *z*-scores of BW and BH increased year by year (Figure 2), although all the mean *z*-scores of BW, BH, and BMI were below the means of WHO standards and the mean *z*-score of BMI was stationary during the last few years. From 2012 to 2018, the annual mean *z*-scores for BW were −1.47, −1.31, −1.24, −1.09, −0.90, −0.86, and −0.74; the serial mean *z*-scores for BH were −1.44, −1.37, −1.20, −1.18, −1.16, −1.10, and −0.92; the serial mean *z*-scores for BMI were −0.85, −0.71, −0.79, −0.67, −0.54, −0.58, and −0.63 (Appendix A).

Figure 3 illustrated this population’s growth curves of BW, BH, and BMI by showing the 3rd, the 15th, the 50th, the 85th, and the 97th percentiles according to the distribution of participants at different ages. The curves generally fell below the 85th percentile of the WHO standards. Moreover, the annual growth rates of height (cm/year) were plotted in Figure 4, and details were provided in Appendix A. In terms of growth rates of height, boys grew the most rapidly at speeds of 7.07 cm/year (from 13 to 14 years old) and 7.02 cm/year (from 14 to 15 years old); girls grew the most rapidly at speeds of 6.52 cm/year (from 10 to 11 years old) and 6.51 cm/year (from 11 to 12 years old).

## 4. Discussion

Our study presented two main results: (1) long-term growth trends of BW, BH, and BMI for children aged 3 to 18 years in a high-altitude area and (2) the growth curves of this high-altitude population. Their mean *z*-scores of BW and BH increased gradually year by year. The growth curves we showed would be informative and helpful in understanding child health and child growth patterns in Ladakh or similar high-altitude areas. Our study is the first cohort study showing the long-term growth trends of highland-living children.

Our utmost strength was that we observed this cohort for an extended period, while most of the previous studies regarding this topic were cross-sectional [5,6,10,11,13,14,15,34]. The cohort design of this study presented three main benefits. First of all, we showed details of long-term growth trends, which were limited in previous studies. Few prior studies observed child growth longitudinally. One Peruvian study followed 300 Quechua people (3000–4000 m) aged from 1 to 22 years for a short period of three years [7]. Another study followed 40 infants in Bolivia (3600 m) every month since birth for one year [10]. Wang et al. also followed a cohort of 253 Tibetan infants (3500 m) for one year after birth [12]. The last two studies only focused on infant populations. Second, long-term observation allowed us to understand more nuances in growth, such as growth spurts when entering puberty. Previous studies showed that highland-living children had ill-defined or delayed growth spurts; however, most of them were early studies in the 1960s and 1970s [7,14,35,36]. In 1978, a study demonstrated that Bods people of Ladakh did not have clear growth spurts and showed their bimodal peaks of height at the age of 13 to 14 years and 15 to 16 years [14]. In contrast to this study, our results showed notable growth spurts of height at the age of 13 to 15 years for boys and 10 to 12 years for girls, but the peak for girls was less prominent. On the other hand, in comparison with other international data, the times of growth spurts in our study were not significantly later, although the growth rates (cm/year) of both genders were slower. One US study showed that white children had peak height velocities at the age of 13 to 14 years for boys (9.56 cm/year) and 11 to 12 years for girls (8.09 cm/year) [37]. However, the interpretation of this phenomenon requires further research. Third, repeated measurements of a cohort can reduce the intra- and inter-individual variations and provide a higher validity of the results. The estimation would be more precise compared to previous small size studies with short-term observations.

We found that this population’s mean *z*-scores of BW and BH increased year by year, while the exact causes remained unclear. Our team suggested increasing milk and egg intake frequency per week to improve students’ nutritional conditions, but its contribution to BW and BH increments was uncertain. Socioeconomic determinants might be possible contributing factors [38,39]. Improved socioeconomic statuses ensure better food security and eliminate malnutrition and the lack of micronutrients [40]. Although some direct and indirect evidence might suggest that socioeconomic conditions improved in Ladakh, the information about the individual socioeconomic status of participants was unavailable in this study [24,39,41]. It is difficult to determine whether their improved socioeconomic conditions positively affected child growth conditions.

Child growth is affected by multiple factors, including intrauterine growth, maternal health, environment, socioeconomic conditions, and resources and infrastructure [6]. Many previous studies have tried to investigate the complex mechanisms of child growth restriction in high-altitude areas [6,9,10,11,34]. For example, chronic hypoxia stress would cause a delay in linear growth because of insufficient oxygen supply for optimal musculoskeletal development [42]. Dang et al. concluded that living in high-altitude areas, independent of socioeconomic factors, led to stunting among Tibetan children [6]. However, Harris et al. reported that the altitude of residence was not the determining factor of growth failure by comparing the nutritional statuses of Tibetan children at three different altitudes [34]. On the other hand, socioeconomic deprivation also played a critical role in child growth restriction in several studies [8,42]. While some studies tried to separate the effects of high altitude and socioeconomic conditions on child growth, living in high-altitude areas, which are often resource-limited, exposes the residents to various risk factors of growth restriction simultaneously [6,34,42].

In addition, a number of studies raised a critical concern that using universal standards to evaluate child growth in diverse areas of the world may result in biased estimates of their growth conditions [16,17,43,44]. Radcliffe et al. indicated that the failure to thrive, in child growth studies, might be caused by the failure to use the right growth charts [16]. Khadilkar et al. showed that even affluent and urban Indian children had suboptimal growth performances based on the WHO Child Growth Standards [43]. A Japanese study found that using the WHO standards would overestimate stunting and underestimate overweight for children in Japan [44]. Some European studies also showed that the WHO standards did not match individual national growth references [45,46,47,48,49]. Because using the universal standards to evaluate child growth in diverse environments or ethnicities might neglect the uniqueness of each population, the academic world has called for region-specific or ethnic group-specific growth curves from either population studies or synthetic methods [19,20,48,50]. Few existing studies focused on developing growth curves for children in high-altitude areas. In our study, the participants had yearly increased mean *z*-scores of BW and BH (Figure 2); the times of growth spurts of body height were similar to international studies (Figure 4); their growth curves generally fell below the 85th percentile of the WHO standards (Figure 3). These all implied that it is crucial to develop growth standards for each child population that take into consideration their unique living environments and possible adaptations to those environments. As governments, nonprofit organizations, and educational institutions have implemented health aid projects for children globally, future research about normal and abnormal growth for each child population will help develop interventions that match their needs and truly benefit them.

Our study has some limitations. First, our study participants were children from a single boarding school in Ladakh, India. External validation would be needed in the future. Second, we did not investigate the exact altitude of each participant’s hometown, although most of them were from villages at altitudes from 2800 to 3700 m. Third, we do not have the individual socioeconomic status of study participants. Future research is warranted to study potential factors influencing child growth in high-altitude areas. Fourth, intrauterine growth is related to child growth after birth; however, the data about the participants’ birth weight were unavailable in this study. Last, the analysis of health checkup results other than body weight and body height was not included. Again, rather than focusing on the effects of high altitudes and other relevant factors in high-altitude areas on child growth, this study aimed to provide a descriptive observation on the long-term child growth pattern in this unique environment.

## 5. Conclusions

Our study is the first cohort study presenting the long-term growth trends of children in high-altitude Ladakh. The study population showed improved body weight and body height year by year; the annual anthropometric measurements were used to depict growth curves, which generally fell below the 85th percentile of the WHO standards. The results are informative and useful for future research on child health and child growth in both Ladakh and other high-altitude areas with similar environmental conditions. The underlying mechanisms of these findings need future research on more representative data of high-altitude populations.

## Figures and Tables

**Figure 1 ijerph-17-03652-f001:**
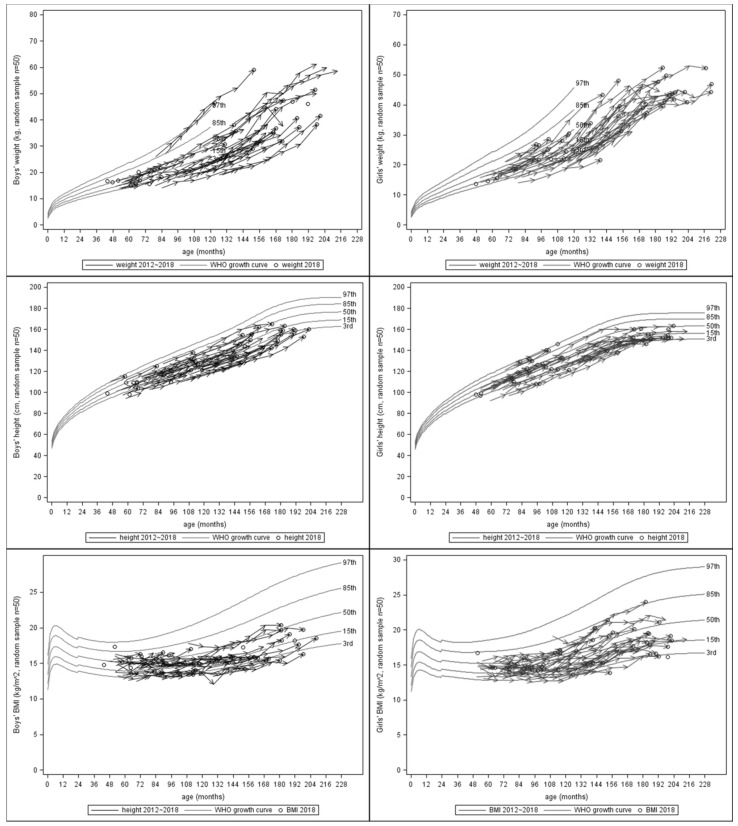
Serial plots of BW, BH, and BMI of randomly selected 50 sampled participants from 2012 to 2018, superimposed on the WHO standards (BW: body weight, BH: body height, BMI: body mass index).

**Figure 2 ijerph-17-03652-f002:**
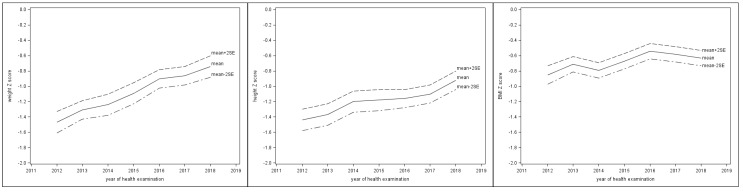
*Z*-score trends of BW, BH, and BMI from 2012 to 2018 superimposed on the WHO standards (BW: body weight, BH: body height, BMI: body mass index).

**Figure 3 ijerph-17-03652-f003:**
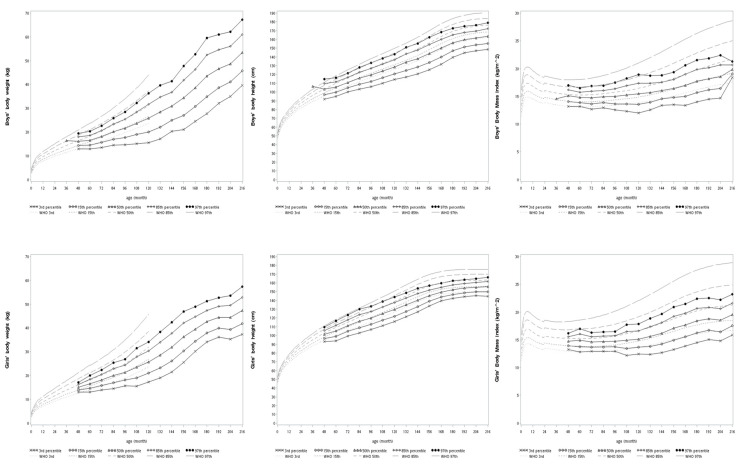
Growth curves of BW, BH, and BMI superimposed on the WHO standards (BW: body weight, BH: body height, BMI: body mass index).

**Figure 4 ijerph-17-03652-f004:**
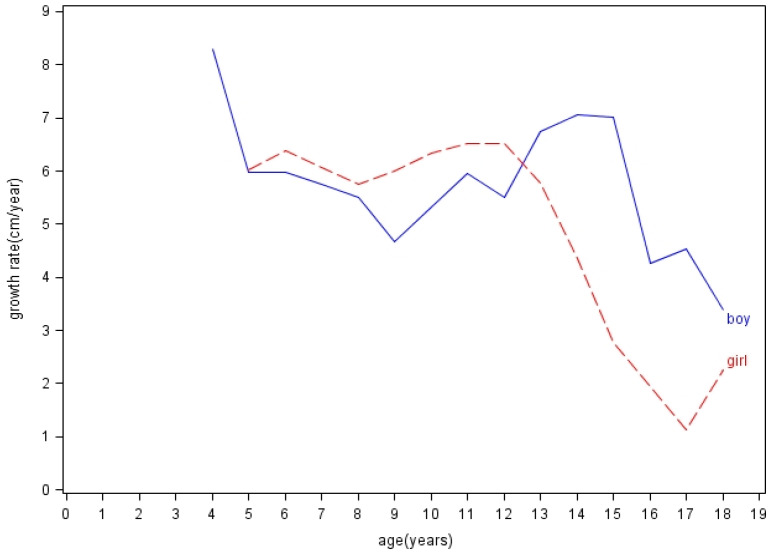
The annual body height growth rates (cm/year) of both genders at different ages.

**Table 1 ijerph-17-03652-t001:** Participant enrollment and loss of follow-up from 2012 to 2018.

	2012	2013	2014	2015	2016	2017	2018
number of enrolled participants	195	216	242	280	290	301	322
number of newly-enrolled participants	195	21	27	45	43	37	50
number of absence of follow-up	N/A	0	1	7	33	26	29
percentage (%) of loss of follow-up	N/A	0%	0.5%	2.9%	11.8%	9.0%	9.6%

**Table 2 ijerph-17-03652-t002:** Body weight, body height, and body mass index of participants stratified by age and gender.

Age, Years	Number (Boy/Girl)	BW, Mean (SE), kg		BH, Mean (SE), cm		BMI, Mean (SE), kg/m^2^	
		Boys	Girls	Boys	Girls	Boys	Girls
3	1 (1/0)	11.5 (0.28)	NA	89.7 (0.65)	NA	15.0 (0.16)	NA
4	42 (19/23)	13.8 (0.28)	12.9 (0.28)	97.9 (0.65)	96.6 (0.67)	14.8 (0.18)	14.6(0.16)
5	96 (54/42)	15.6 (0.25)	14.6 (0.27)	103.8(0.58)	102.6(0.60)	14.7 (0.11)	14.5(0.15)
6	149 (76/73)	17.4 (0.23)	16.5 (0.28)	109.4(0.51)	108.8(0.55)	14.7 (0.10)	14.3(0.11)
7	171 (93/78)	19.7 (0.24)	18.6 (0.29)	114.9(0.52)	114.5(0.54)	14.9 (0.09)	14.4(0.11)
8	183 (102/81)	21.7 (0.27)	20.7 (0.31)	120.3(0.56)	119.8(0.57)	14.9(0.11)	14.5(0.11)
9	197 (110/87)	24.0 (0.33)	23.3 (0.34)	124.9(0.56)	125.5(0.57)	15.3(0.12)	14.8(0.12)
10	200 (108/92)	26.7 (0.43)	26.4 (0.40)	130.2(0.57)	131.1(0.63)	15.6(0.15)	15.2(0.13)
11	194 (105/89)	29.8 (0.48)	30.2 (0.48)	136.1(0.66)	137.6(0.67)	15.9(0.14)	15.8(0.15)
12	171 (84/87)	32.9 (0.54)	34.3 (0.53)	141.4(0.75)	143.9(0.68)	16.3(0.13)	16.5(0.17)
13	145 (69/76)	36.8 (0.68)	38.8 (0.56)	147.8(0.89)	149.5(0.65)	16.7(0.15)	17.5(0.19)
14	117 (56/61)	41.2 (0.81)	42.5 (0.57)	154.6(0.94)	153.7(0.67)	17.2(0.21)	18.2(0.21)
15	85 (33/52)	46.4 (0.88)	45.5 (0.58)	161.3(1.05)	156.3(0.73)	17.9(0.22)	18.8(0.22)
16	56 (21/35)	50.1 (1.09)	47.6 (0.64)	165.7(1.43)	158.1(0.77)	18.4(0.27)	19.3(0.24)
17	29 (10/19)	54.4 (1.24)	48.6 (0.72)	170.1(1.93)	159.1(0.79)	19.1(0.33)	19.4(0.30)
18	11 (2/9)	58.1 (1.93)	50.2 (0.83)	173.6(4.03)	161.0(1.05)	19.8(0.35)	19.6(0.32)

SE: standard error, BW: body weight (kg), BH: body height (cm), BMI: body mass index (kg/m^2^). Means (SE) were estimated by generalized estimating equation (GEE) for repeated measurements.

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
