# Peer review of "Child Growth Curves in High-Altitude Ladakh: Results from a Cohort Study"

_ijerph, 2020, doi:10.3390/ijerph17103652_

Round 1

Reviewer 1 Report

Abstract

The authors may include body weight and body height in keywords

Introduction

lINE#34 avoid using etc., in the introduction that creates confusion for the readers.   

Material and methods

Line 62 Correct the word of India 

Line #90  skin, and past history and menstruation history for girls 

What do you mean past history here? Do you want to link with menstruation past history?

Line#91 for the vision exam

You may write vision examination or eye health 

Line#89 and 90

What is the objective of mentioning these below services if you are not included all these variables in your analysis??

The volunteer physicians performed physical examinations covering ear, throat, heart, chest, abdomen, skin, and past history and menstruation history for girls.

Results:

My major concern with the result section is why the authors did not include the socio-economic and background characteristics of children. The authors may better conclude their results if they include some background characteristics of the children.

 Discussion

The authors may also discuss the child growth standard with previous other studies and can conclude their results with reference to child growth and reference standard data.

According to your results mean Z scores of BW, BH, and BMI
26 were -1.47, -1.44, and -0.85 in 2012 and increased to -0.74, -0.92, and -0.63 in 2018.

What are the possible reasons these outcomes got better? Can you discuss in detail the discussion section about this positive progress align with previous studies? 

The result of fig 04 needs to discuss into the discussion section.  

Conclusions

Need to revise and strengthen this section. You may quickly summarize your key messages into this section 

Author Response

Response to Reviewer 1 Comments

Point 1: Abstract: The authors may include body weight and body height in keywords

Response 1: Thank you so much for this advice. We have included “body weight” and “body height” in keywords based on your suggestion. (Line#31)

Point 2: Introduction: line#34 avoid using etc., in the introduction that creates confusion for the readers.   

Response 2: Thank you for this comment. We have edited it based on your suggestion. Line#34-36:

Low-oxygen environments, scarceness of resources, and lack of health infrastructure are common risk factors of poor health and restricted growth.

Point 3: Line 62: Correct the word of India 

Response 3: We sincerely apologize for this typo and have corrected the word. (Line#70)

Point 4: Line #90:  skin, and past history and menstruation history for girls. What do you mean past history here? Do you want to link with menstruation past history?

Response 4: Thank you for pointing this out. We actually mean the participants’ “general medical past history,” which refers to whether they have major diseases or surgical history or not. We should have explained it clearly and have edited it based on your suggestion.

Line#97-98:

The volunteer physicians performed physical examinations covering ear, throat, heart, chest, abdomen, skin, past medical history, and menstruation history for girls.

Point 5: Line#91 for the vision exam. You may write vision examination or eye health 

Response 5: Thank you for the suggestion. We have corrected “vision exam” to “eye health” accordingly. (Line#99)

Point 6: Line#89, and 90. What is the objective of mentioning these below services if you are not included all these variables in your analysis?? The volunteer physicians performed physical examinations covering ear, throat, heart, chest, abdomen, skin, and past history and menstruation history for girls.

Response 6: Since this study is mainly focused on the participants’ body weight, body height, BMI, and the development of their growth curves, we decided not to include the analysis of checkup results to save more space for relevant discussions. We did provide the results of health checkups to our partner school for further intervention. We believe that providing a comprehensive description of our health checkup can give a full picture of what we have done so that readers can understand this study thoroughly.

Point 7: Results: My major concern with the result section is why the authors did not include the socio-economic and background characteristics of children. The authors may better conclude their results if they include some background characteristics of the children.

Response 7: Thank you so much for pointing this out. We understood that this is a major limitation of this study and mentioned it in Line#290-292. When designing the study, we realized the difficulties of gathering the socioeconomic data since a lot of our participants were from remote mountain areas, and some of them were orphans. It was impractical and impossible to collect information about the participants’ individual socioeconomic background. Therefore, we tried our best to find surrogate information about the local general socioeconomic condition. (Line#74-78, Line#251-253) We believe that a long-term descriptive study like ours can provide valuable information about the growth trend of this high-altitude living population, though the limitation does exist.

Point 8: Discussion. The authors may also discuss the child growth standard with previous other studies and can conclude their results with reference to child growth and reference standard data.

Response 8: Yes, we discussed other studies which used the WHO standards as references in Line#271-275. Many studies have found that using the WHO standards to evaluate child growth in different areas would overestimate or underestimate their growth. In Line#271-273, an Indian study showed that even urban Indian children had suboptimal growth conditions when using the WHO standards to evaluate. In Line#273-274, we mentioned that a Japanese study showed that using the WHO standards overestimated stunting and underestimated overweight among Japanese children; in Line#274-275, some European studies also showed that the WHO standards did not match individual national growth references. Therefore, there is a call for region-specific or ethnic group-specific child growth curves. This is the main reason that we want to build up growth curves for this underserved high-altitude living child population.

Point 9: According to your results mean Z scores of BW, BH, and BMI
26 were -1.47, -1.44, and -0.85 in 2012 and increased to -0.74, -0.92, and -0.63 in 2018. What are the possible reasons these outcomes got better? Can you discuss in detail the discussion section about this positive progress align with previous studies? 

Response 9: Thank you for this comment. We had relevant discussions in Line#246-253. As we mentioned in the manuscript, the exact reasons behind these outcomes remained unclear. We proposed possible explanations, and further research is warranted.

Point 10: The result of fig 04 needs to discuss into the discussion section. 

Response 10: Thank you for this comment. We provided relevant discussions in Line# 236-242. Our results showed that both genders had the most rapid growth rates of height at certain age periods, while most of the previous studies showed ill-defined growth spurts. We also compared our results with a more recent study in Line#238-242.

Point 11: Conclusions. Need to revise and strengthen this section. You may quickly summarize your key messages into this section 

Response 11: Thank you for this suggestion. We have reviewed the section thoroughly and edited accordingly, summarizing the main findings of our study in the conclusion paragraph.

Line#300-320:

Our study is the first cohort study presenting the long-term growth trends of children in high-altitude Ladakh. The study population showed improved body weight and body height year by year; the annual anthropometric measurements were used to depict growth curves, which generally fell below the 85th percentile of the WHO standards. The results are informative and useful for future research on child health and child growth in both Ladakh and other high-altitude areas with similar environmental conditions. The underlying mechanisms of these findings need future research on more representative data of high-altitude populations.

Reviewer 2 Report

This is a good and very useful study and provide many valuable information. This study shows growth pattern is tallying with WHO standards. This study population is a deprived group and not a normal population, it is expected to grow low level than WHO standard. But it shows with the time there was a tremendous improvement of the growth. Though the height velocity was 1.5-2 cm below than the optimum, it goes beyond the expected ages showing extended period of growth. This is good picture of this study population for catch up growth. Authors need to explain how did they randomly select 50 participants. But the results were not discussed properly by the authors.

Line 28 – 29

“This is the first cohort study about long-term child growth patterns in a high-altitude area”. There is a cohort in Tibet for infants. This statement is not a good conclusion for this study. May be something like “need more representative data of high-altitude population and i- depth studies”

Line 68

Need correction – India

Line 77-80

This section shows these children have so many underlying causes of malnutrition which affect their usual growth. Need to identify this in the discussion section.

Line 82-98

According to this section health data is available, but not presented or even comment about it. This is a limitation in this study. Health status directly related to the nutritional status as an immediate cause. If the health status of the population is poor which will affect weight and height. This should be reflected in the discussion.

Line 103-104

In this paper wasting, stunting and underweight are not shown in the results section. Authors can remove it, if they are not going to present data, if needed WHO definition is needed for different ages.

Line 124

“Dynamic cohort” – not properly explained, if not can remove the word dynamic.

Line 125-126

It is good if authors provide data on (number of children) left due to personal issues, graduated.

Line 154-155 – Table 2

Need a new column with number of children in each age group.

Line 214-215

This is the place authors should discuss about underlying causes of these children who are coming from deprived population.

Line 242-245

Authors should discuss the height velocity. It looks very positive pattern.

Line 251-258

One of the limitations is health status and nutrition intake (special diet) of the population is not presented and birth weight of the children. Adequate Intrauterine growth is also important for optimum growth. Need to add as limitations.

Line 260-263

Need in-depth study in a more representative population to identify the pattern of growth to understand the growth pattern is similar or different than WHO standard.

Author Response

Response to Reviewer 2 Comments

Point 1: This is a good and very useful study and provide many valuable information. This study shows growth pattern is tallying with WHO standards. This study population is a deprived group and not a normal population, it is expected to grow low level than WHO standard. But it shows with the time there was a tremendous improvement of the growth. Though the height velocity was 1.5-2 cm below than the optimum, it goes beyond the expected ages showing extended period of growth. This is good picture of this study population for catch up growth. Authors need to explain how did they randomly select 50 participants. But the results were not discussed properly by the authors.

Response 1: We appreciate this comment very much. We do hope that our study can provide valuable information about the growth tread of a population in such a resource-constrained area. About the question regarding the sampling, we should have clarified that only Figure 1 was depicted by sampling 50 cases. If all participants were depicted in Figure 1, all the lines would make it very difficult to read. Also, based on your suggestion, we have explained how we sampled in Line#144-147. To clarify, all of the rest analyses were done using the data of all participants, not the 50 samples.

Line#144-147:

The serial BW, BH, and BMI of 50 random samples, which were selected with equal probability without replacement from the original population, were presented superimposed on the WHO standards in Figure 1.

Point 2: Line 28 – 29. “This is the first cohort study about long-term child growth patterns in a high-altitude area”. There is a cohort in Tibet for infants. This statement is not a good conclusion for this study. May be something like “need more representative data of high-altitude population and i- depth studies”

Response 2: Thank you for this comment. We understand that there were some other cohort studies like the one you mentioned, but none of them were long-term observatory studies like ours, which we believe is the most valuable point of our study. Also, because of this study’s limitations, more representative data of high-altitude population will be needed in the future. We have edited it based on your suggestion.

Line#29-30:

The detailed underlying mechanisms of our findings need future research on more representative data of high-altitude populations.

Point 3: Line 68 Need correction – India

Response 3: We sincerely apologize for this typo and have corrected the word. (Line#70)

Point 4: Line 77-80. This section shows these children have so many underlying causes of malnutrition which affect their usual growth. Need to identify this in the discussion section.

Response 4: Thank you for this comment. We agree with you that any underlying factors can affect child growth in high-altitude areas. Based on your suggestion, we revised a paragraph in the discussion (Line#255-267) and explained possible reasons, including low oxygen in mountains, socioeconomic factors, etc.

Line#255-267:

Child growth is affected by multiple factors, including intrauterine growth, maternal health, environment, socioeconomic conditions, and resource and infrastructure [6]. Many previous studies have tried to investigate the complex mechanisms of child growth restriction in high-altitude areas [6,9-11,34]. For example, chronic hypoxia stress would cause a delay in linear growth because of insufficient oxygen supply for optimal musculoskeletal development [42]. Dang et al. concluded that living in high-altitude areas, independent of socioeconomic factors, led to stunting among Tibetan children [6]. However, Harris et al. reported that the altitude of residence was not the determining factor of growth failure by comparing the nutritional statuses of Tibetan children at three different altitudes [34]. On the other hand, socioeconomic deprivation also played a critical role in child growth restriction in several studies [8, 42]. While some studies tried to separate the effects of high altitude and socioeconomic conditions on child growth, living in high-altitude areas, which are often resources-limited, exposes the residents to various risk factors of growth restriction simultaneously [6,34,42].

Point 5: Line 82-98. According to this section health data is available, but not presented or even comment about it. This is a limitation in this study. Health status directly related to the nutritional status as an immediate cause. If the health status of the population is poor which will affect weight and height. This should be reflected in the discussion.

Response 5: We appreciate this comment very much. However, the analysis of health checkup results was unavailable in this manuscript. Our health checkup included the examination of eye, ear, throat, heart, chest, abdomen, and skin, which were difficult to quantify and present this manuscript. Accordingly, we incorporated this point as one of our limitations. (Line#294-295)

Line# 294-295:

Last, the analysis of health checkup results other than body weight and body height was not included.

Point 6: Line 103-104 In this paper wasting, stunting and underweight are not shown in the results section. Authors can remove it, if they are not going to present data, if needed WHO definition is needed for different ages.

Response 6: Yes, we provided the results about wasting, stunting, and underweight in the result section Line#147-156. The percentages of participants who were stunting and underweight decreased year by year from 2012 to 2018. Also, the results of Z score trends were presented in Line#156-161. Figure 2 in Page 6 also showed the improved Z score trends of BW, BH, and BMI. Therefore, the discussion of wasting, stunting, and underweight is necessary for this study.

Point 7: Line 124. “Dynamic cohort” – not properly explained, if not can remove the word dynamic.

Response 7: Thank you for this comment. We explained why this was a dynamic cohort in Line#134-135. Because new students would enter the cohort annually and some might leave due to graduation or personal reasons, this cohort was dynamic. To avoid confusion, we deleted the sentence of “This cohort was dynamic.” as you suggested. (Line#134)

Point 8: Line 125-126. It is good if authors provide data on (number of children) left due to personal issues, graduated.

Response 8: Thank you for pointing this out. Table 1 (Page 4) showed that our study’s annual loss of follow-up rates were low, except for 2018, the year when few participants left the cohort because of graduation.

Point 9: Line 154-155 – Table 2. Need a new column with number of children in each age group.

Response 9: We very much appreciate this comment and agree that we should provide the number to give comprehensive information. We have edited Table 2 and provided the data. (Page 4)

The revised Table 2 is as follows:

Age, years

Number (boy/girl)

BW, mean (SE), kg

BH, mean (SE), cm

BMI, mean (SE), kg/m2

Boys

Girls

Boys

Girls

Boys

Girls

3

1 (1/0)

11.5 (0.28)

NA

89.7 (0.65)

NA

15.0 (0.16)

NA

4

42 (19/23)

13.8 (0.28)

12.9 (0.28)

97.9 (0.65)

96.6 (0.67)

14.8 (0.18)

14.6(0.16)

5

96 (54/42)

15.6 (0.25)

14.6 (0.27)

103.8(0.58)

102.6(0.60)

14.7 (0.11)

14.5(0.15)

6

149 (76/73)

17.4 (0.23)

16.5 (0.28)

109.4(0.51)

108.8(0.55)

14.7 (0.10)

14.3(0.11)

7

171 (93/78)

19.7 (0.24)

18.6 (0.29)

114.9(0.52)

114.5(0.54)

14.9 (0.09)

14.4(0.11)

8

183 (102/81)

21.7 (0.27)

20.7 (0.31)

120.3(0.56)

119.8(0.57)

14.9(0.11)

14.5(0.11)

9

197 (110/87)

24.0 (0.33)

23.3 (0.34)

124.9(0.56)

125.5(0.57)

15.3(0.12)

14.8(0.12)

10

200 (108/92)

26.7 (0.43)

26.4 (0.40)

130.2(0.57)

131.1(0.63)

15.6(0.15)

15.2(0.13)

11

194 (105/89)

29.8 (0.48)

30.2 (0.48)

136.1(0.66)

137.6(0.67)

15.9(0.14)

15.8(0.15)

12

171 (84/87)

32.9 (0.54)

34.3 (0.53)

141.4(0.75)

143.9(0.68)

16.3(0.13)

16.5(0.17)

13

145 (69/76)

36.8 (0.68)

38.8 (0.56)

147.8(0.89)

149.5(0.65)

16.7(0.15)

17.5(0.19)

14

117 (56/61)

41.2 (0.81)

42.5 (0.57)

154.6(0.94)

153.7(0.67)

17.2(0.21)

18.2(0.21)

15

85 (33/52)

46.4 (0.88)

45.5 (0.58)

161.3(1.05)

156.3(0.73)

17.9(0.22)

18.8(0.22)

16

56 (21/35)

50.1 (1.09)

47.6 (0.64)

165.7(1.43)

158.1(0.77)

18.4(0.27)

19.3(0.24)

17

29 (10/19)

54.4 (1.24)

48.6 (0.72)

170.1(1.93)

159.1(0.79)

19.1(0.33)

19.4(0.30)

18

11 (2/9)

58.1 (1.93)

50.2 (0.83)

173.6(4.03)

161.0(1.05)

19.8(0.35)

19.6(0.32)

Point 10: Line 214-215 This is the place authors should discuss about underlying causes of these children who are coming from deprived population.

Response 10: We appreciate this comment. The underlying causes of a high-altitude population being growth-restricted are complex. To provide more discussion, we revised the paragraph (Line#255-267) and explained possible reasons, including low oxygen in mountains, socioeconomic factors, etc.

Line#255-267:

Child growth is affected by multiple factors, including intrauterine growth, maternal health, environment, socioeconomic conditions, and resource and infrastructure [6]. Many previous studies have tried to investigate the complex mechanisms of child growth restriction in high-altitude areas [6,9-11,34]. For example, chronic hypoxia stress would cause a delay in linear growth because of insufficient oxygen supply for optimal musculoskeletal development [42]. Dang et al. concluded that living in high-altitude areas, independent of socioeconomic factors, led to stunting among Tibetan children [6]. However, Harris et al. reported that the altitude of residence was not the determining factor of growth failure by comparing the nutritional statuses of Tibetan children at three different altitudes [34]. On the other hand, socioeconomic deprivation also played a critical role in child growth restriction in several studies [8, 42]. While some studies tried to separate the effects of high altitude and socioeconomic conditions on child growth, living in high-altitude areas, which are often resources-limited, exposes the residents to various risk factors of growth restriction simultaneously [6,34,42].

Point 11: Line 242-245. Authors should discuss the height velocity. It looks very positive pattern.

Response 11: Thank you for this comment. We provided relevant discussions in Line# 235-242. Our results showed that both genders had the most rapid growth rates of height at certain age periods, while most of the previous studies showed ill-defined growth spurts. We also compared our results with a more recent study in Line#238-242.

Point 12: Line 251-258. One of the limitations is health status and nutrition intake (special diet) of the population is not presented and birth weight of the children. Adequate Intrauterine growth is also important for optimum growth. Need to add as limitations.

Response 12: Thank you for pointing this out. We agree that the intrauterine growth condition is important and closely related to growth conditions after birth. However, it was impossible to collect the relevant information since almost all of our participants lived away from their families. As you suggested, we mentioned the importance of intrauterine growth and also incorporated this point as our limitation. (Line#255-256, Line#293-294)

Line#255-256:

Child growth is affected by multiple factors, including intrauterine growth, maternal health, environment, socioeconomic conditions, and resource and infrastructure.

Line#293-294:

Fourth, intrauterine growth is related to child growth after birth; however, the data about the participants’ birth weight was unavailable in this study.

Point 13: Line 260-263 Need in-depth study in a more representative population to identify the pattern of growth to understand the growth pattern is similar or different than WHO standard.

Response 13: We really appreciate this comment. Though our study population was relatively small, the repeated measurement in our long-term follow-up could reduce the intra- and inter-individual variations and provide a higher validity of the results. (Line#243-244). We agree that future research on a more representative population is needed, and we edited the conclusion accordingly. (Line#319-320)

Line#319-320:

The underlying mechanisms of these findings need future research on more representative data of high-altitude populations.

Reviewer 3 Report

This manuscript addresses an important child health isuues in high altitude areas, reporting informative results. My comments are as follows:

  1. With regard to the title, the subtitle may precisely include “ Results from a cohort study of a charity private institution”. Because, the study population is children of a charity private school.
  2. Page 4; Readers will want to know the number of participants by age and gender in the Table 2. Are the numbers shown in the Table S2 ?

Please confirm “Author’s guideline” on the reference (e.g. page 9 line 307; ......etc....)

Author Response

Response to Reviewer 3 Comments

Reviewer 3: This manuscript addresses an important child health issues in high altitude areas, reporting informative results. My comments are as follows:

Point 1: With regard to the title, the subtitle may precisely include “ Results from a cohort study of a charity private institution”. Because, the study population is children of a charity private school.

Response 1: Thank you for this comment. In Line#80-82, we provided the information about the study participants and mentioned all of them were from our partner school, a private charity institution. Since many of our participants were from remote mountain areas in Ladakh, we believe that our original title will better represent their high-altitude living environment.

Point 2: Page 4; Readers will want to know the number of participants by age and gender in the Table 2. Are the numbers shown in the Table S2 ?

Response 2: We very much appreciate this comment and agree that we should provide the number to give comprehensive information. Based on your suggestion, we have edited Table 2 and added a column of the number of participants by age and gender. Please refer to Page 4. The revised Table 2 is as follows:

Age, years

Number (boy/girl)

BW, mean (SE), kg

BH, mean (SE), cm

BMI, mean (SE), kg/m2

Boys

Girls

Boys

Girls

Boys

Girls

3

1 (1/0)

11.5 (0.28)

NA

89.7 (0.65)

NA

15.0 (0.16)

NA

4

42 (19/23)

13.8 (0.28)

12.9 (0.28)

97.9 (0.65)

96.6 (0.67)

14.8 (0.18)

14.6(0.16)

5

96 (54/42)

15.6 (0.25)

14.6 (0.27)

103.8(0.58)

102.6(0.60)

14.7 (0.11)

14.5(0.15)

6

149 (76/73)

17.4 (0.23)

16.5 (0.28)

109.4(0.51)

108.8(0.55)

14.7 (0.10)

14.3(0.11)

7

171 (93/78)

19.7 (0.24)

18.6 (0.29)

114.9(0.52)

114.5(0.54)

14.9 (0.09)

14.4(0.11)

8

183 (102/81)

21.7 (0.27)

20.7 (0.31)

120.3(0.56)

119.8(0.57)

14.9(0.11)

14.5(0.11)

9

197 (110/87)

24.0 (0.33)

23.3 (0.34)

124.9(0.56)

125.5(0.57)

15.3(0.12)

14.8(0.12)

10

200 (108/92)

26.7 (0.43)

26.4 (0.40)

130.2(0.57)

131.1(0.63)

15.6(0.15)

15.2(0.13)

11

194 (105/89)

29.8 (0.48)

30.2 (0.48)

136.1(0.66)

137.6(0.67)

15.9(0.14)

15.8(0.15)

12

171 (84/87)

32.9 (0.54)

34.3 (0.53)

141.4(0.75)

143.9(0.68)

16.3(0.13)

16.5(0.17)

13

145 (69/76)

36.8 (0.68)

38.8 (0.56)

147.8(0.89)

149.5(0.65)

16.7(0.15)

17.5(0.19)

14

117 (56/61)

41.2 (0.81)

42.5 (0.57)

154.6(0.94)

153.7(0.67)

17.2(0.21)

18.2(0.21)

15

85 (33/52)

46.4 (0.88)

45.5 (0.58)

161.3(1.05)

156.3(0.73)

17.9(0.22)

18.8(0.22)

16

56 (21/35)

50.1 (1.09)

47.6 (0.64)

165.7(1.43)

158.1(0.77)

18.4(0.27)

19.3(0.24)

17

29 (10/19)

54.4 (1.24)

48.6 (0.72)

170.1(1.93)

159.1(0.79)

19.1(0.33)

19.4(0.30)

18

11 (2/9)

58.1 (1.93)

50.2 (0.83)

173.6(4.03)

161.0(1.05)

19.8(0.35)

19.6(0.32)

Point 3: Please confirm “Author’s guideline” on the reference (e.g. page 9 line 307; ......etc....)

Response 3: Thank you for pointing this out. We are worry for the error. We have edited the reference part based on the guideline. (https://mdpi-res.com/data/mdpi_references_guide_v5.pdf)

Please refer to Line#346-347, 362, 371, 380, 388, 426, 432, 434, 449, 457, 460-461, 463-464.

Reviewer 4 Report

Thank you for a very nice manuscript.

Here are some suggestions you could consider:

Introduction: The first paragraph has some redundancies and could be improved.

Line 33- you mention "various mechanisms", I suggest you mention some of them. 

Line 34-38: It is not very clear, I suggest you clarify this part. For example instead of merging resource scarcity and environmental factors, better to show how each relates to the health and nutrition problem.

Materials and Methods section:

Line 62: India (change the last letter to small letter)

Line 91: It would be good to include some information about the training, whether or not there was standardization done? 

Results: my suggestion for this section is to check the quality of the figures, it is difficult to see the lines.

Author Response

Response to Reviewer 4 Comments

Thank you for a very nice manuscript. Here are some suggestions you could consider:

Point1: Introduction: The first paragraph has some redundancies and could be improved.

Response 1: We really appreciate this comment. Based on your suggestion, we have reviewed our first paragraph thoroughly and cut some redundancies to make it more precise and easier to read.(Line#34-39)

Line#34-39:

Living in high-altitude areas affects child health through various mechanisms. Low-oxygen environments, scarceness of resources, and lack of health infrastructure are common risk factors of poor health and restricted growth [1]. Children are more likely to have respiratory infectious diseases, poor nutritious condition, and growth retardation [2,3]. Some of these risk factors are inherently associated. Although there is no single leading risk factor, the impact of living in high-altitude areas on child health is certain.

Point 2: Line 33- you mention "various mechanisms", I suggest you mention some of them. 

Response 2: Yes, we mentioned some mechanisms in Line#34-35. Low-oxygen environments, resource scarcity, and lack of healthcare infrastructure are three of the most commonly documented driving factors that influence child health in high-altitude areas. Relevant discussion was also mentioned in Line#255-267.

Point 3: Line 34-38: It is not very clear, I suggest you clarify this part. For example instead of merging resource scarcity and environmental factors, better to show how each relates to the health and nutrition problem.

Response 3: Thank you so much for this comment. When reviewing literature, we found that some studies tried to separate the effects of high-altitude environment and resource scarcity on child health. However, some studies admitted that it was difficult to tell them apart since. In order to make clear discussion, we revised a paragraph in the discussion based on your suggestion. (Line#255-267)

Line#255-267:

Child growth is affected by multiple factors, including intrauterine growth, maternal health, environment, socioeconomic conditions, and resource and infrastructure [6]. Many previous studies have tried to investigate the complex mechanisms of child growth restriction in high-altitude areas [6,9-11,34]. For example, chronic hypoxia stress would cause a delay in linear growth because of insufficient oxygen supply for optimal musculoskeletal development [42]. Dang et al. concluded that living in high-altitude areas, independent of socioeconomic factors, led to stunting among Tibetan children [6]. However, Harris et al. reported that the altitude of residence was not the determining factor of growth failure by comparing the nutritional statuses of Tibetan children at three different altitudes [34]. On the other hand, socioeconomic deprivation also played a critical role in child growth restriction in several studies [8, 42]. While some studies tried to separate the effects of high altitude and socioeconomic conditions on child growth, living in high-altitude areas, which are often resources-limited, exposes the residents to various risk factors of growth restriction simultaneously [6,34,42].

Point 4: Line 62: India (change the last letter to small letter)

Response 4: We sincerely apologize for this typo and have corrected the word. (Line#70)

Point 5: Line 91: It would be good to include some information about the training, whether or not there was standardization done? 

Response 5: Thank you for this comment. In this study, our volunteers were trained through standardized courses. We have edited it based on your suggestion in Line#98-99.

All volunteers were well-trained through standardized courses and responsible for the vision examination, color blindness test, and anthropometric measurements

Point 6: Results: my suggestion for this section is to check the quality of the figures, it is difficult to see the lines.

Response 6: Thank you for pointing this out. We also noticed that the figures were not as clear as we expected; we were very sorry about this. We have made new figures with higher resolutions and better quality. Please refer to Page 5-7.
